# Head-Only Stunning of Turkeys Part 1: The Minimum Voltage Necessary to Break Down the Inherent High Resistance

**DOI:** 10.3390/ani10122427

**Published:** 2020-12-18

**Authors:** Steve Wotton, Andrew Grist, Mike O’Callaghan, Ed van Klink

**Affiliations:** Bristol Veterinary School, University of Bristol, Bristol BS40 5DU, UK; Steve.Wotton@bristol.ac.uk (S.W.); Andy.grist@bristol.ac.uk (A.G.); mike31@btinternet.com (M.O.)

**Keywords:** turkeys, electrical stunning, impedance

## Abstract

**Simple Summary:**

Pre-slaughter stunning is required for the humane slaughter of turkeys. For the head-only electrical stunning to be effective, the impedance (resistance) in the tissue of the head of the animal between the two electrodes must be overcome by the level of the voltage used. We have assessed the most appropriate voltage to effectively overcome the impedance and provide an effective stun, and that is also safe for the operator to use. Sinusoidal AC at 150 V and 50 Hz was considered to fulfil those requirements and was used for further testing.

**Abstract:**

Pre-slaughter stunning is required for humane slaughter. For turkeys, head-only electrical stunning is most often used by small scale producers. To ensure immediate and effective stunning, the impedance (resistance) of the tissue of the head of the animal situated between the two electrodes needs to be overcome swiftly. The impedance is a function of the voltage and decreases non-linearly with increasing voltage. In this paper, we describe a method to assess the minimum voltage needed at which the impedance no longer decreases, that is likely to produce an effective stun. For ethical reasons, gas stunned, electrically naïve turkeys were used to measure impedance at various levels of voltage and current. Several combinations of voltage and frequency, alternate current (AC), direct current (DC) and pulsed DC, were identified that would be sufficient to achieve the maximum decrease in the impedance, and therefore would allow the highest current and the most effective stun. A minimum, expressed as Root Mean Squared voltage, of 150 V and 50 Hz. would be required in AC, 175 V in pulsed DC at 30% cycle (150 at 50% cycle), and 225 V if voltage spikes of very short duration were used. Sinusoidal AC applied at 150 V, 50 Hz was selected for further testing.

## 1. Introduction

On average, 1.7 to 2 million turkeys are slaughtered in the United Kingdom per month, varying between one million and 2.5 million [1]. Turkey slaughter practices vary, are often seasonal and of low throughput, but this kind of slaughter is not exempt from current welfare regulations [2]. In the UK, the regulations are implemented in the Welfare of Animals at the Time of Killing Regulations 2015 [3]. Pre-slaughter stunning of both livestock and poultry is an essential welfare requirement for humane slaughter [4]. Except for the practice of religious slaughter, EC Regulation 1099/2009 requires the application of an effective and humane stunning method before turkeys are slaughtered. Electrical stunning is one of the stunning methods used in poultry, pigs and sheep [5,6]. In turkeys, a non-penetrative captive bolt [7] as well as gas stunning [8,9] are also used, with gas stunning, normally consisting of gas mixtures containing high concentrations of CO_2_, being far more common than any of the other methods (97% in 2018, 9). When applying electrical stunning, two methods are commonly used for stunning turkeys. One uses a mains-powered hand-held device which applies current across the bird’s head whilst it is restrained in a bleeding cone or shackle. The other method passes the bird’s head through an electric water bath while suspended from a shackle. The water bath stunning method (head-to-body) is more often used, while head only stunning is used by small-scale producers [9,10].

When current is applied across the head, it prevents the direct muscle stimulation produced when current is applied through the whole bird, as happens when a water bath stunner is used. This is likely to reduce damage to the carcass [11] and improve meat quality.

A minimum current threshold level is needed to make sure that a stun is effective, i.e., that an epileptiform insult is being provoked rendering the animal unconscious; where the relationship between current, voltage and resistance is governed by the formula of Ohm’s Law (Current (I) = Voltage (V)/Resistance (R)), this is not straightforward in animal tissue: the impedance of the tissues through which the current is led, is highly influential [5]. Previous research has shown that the impedance of living tissue (pig’s head) was found to be predominantly a function of the stunning voltage used and decreased non-linearly with increasing voltage [12]. The impedance is a main determinant in the immediacy, and hence effectiveness, of the stun: the lower the impedance the more likely the stun will be effective [13,14]. Effectiveness of stunning is defined as initial neck stiffness, absence of rhythmic breathing, absence of spontaneous blinking and third eyelid reflex, absence of other eye reflexes, eyes open and fixed, initial tonic phase, followed by a clonic phase [15]. If the applied voltage is not sufficient to overcome the impedance swiftly, birds will be subjected to painful induction and/or ineffective stunning. The measured impedance is to a certain extent inversely proportional to the voltage used to measure it and the reduction in the impedance with increasing voltage levels off beyond a threshold voltage [12]. This would mean that a minimum voltage level is required to achieve an effective stun, based on electrical parameters, which can be identified.

In this study we seek to determine the minimum voltage needed to break down the inherent high resistance of living tissue i.e., the turkey head, by application of alternating (AC) and direct current (DC) stunning using hand-held electrodes head-only. With this research, we hope to improve the effectiveness of hand-held stunning of turkeys and enable the industry to reduce the variability seen with current practice.

## 2. Materials and Methods

The experiments carried out in this study were conducted under the following ethical approval licences: Home Office Project Licence Number is PPL 30/2438 Stunning, Slaughter and Killing of Poultry Species and Personal Licence Number is PIL 30/1362. Both licences are held by Dr. M. Raj, a former researcher at our institute, who at the time of the experiments was still involved in the work.

A prototype mains-powered generator was designed and built for the purpose of this study. The device could produce a constant voltage pulsed DC that would enable pulse width, duty cycle, applied voltage, and pulse recurring frequency (prf) to be controlled. The specifications of the device are given in Table 1.

Studies on sheep, calves and pigs [12] have shown that the impedance of tissues covering the head remains unchanged for up to 4 h post mortem and therefore the effect of increasing voltage can be measured reliably. Therefore, the experiment could be carried out on gas killed, i.e., electrically naïve turkeys, to avoid the application of low voltages on conscious birds.

### Voltage was Applied as Either

AC and calibrated as Root Mean Squared (RMS, 0.707 × peak voltage) voltage from 50 to 300 volts (V) in 25 V steps with a frequency of 50 Hz.AC and calibrated as Root Mean Squared (RMS, 0.707 × peak voltage) voltage from 50 to 300 V in 25 V steps with a frequency of 200 Hz.Pulsed DC, calibrated by peak voltage and applied from 50 to 300 V in 25 V steps, at 50 Hz and 30% duty cyclePulsed DC, calibrated by peak voltage and applied from 50 to 300 V in 25 V steps, at 50 Hz and 50% duty cyclePulsed DC, calibrated by peak voltage and applied from 50 to 300 V in 25 V steps, at 200 Hz and 30% duty cyclePulsed DC, calibrated by peak voltage and applied from 50 to 300 V in 25 V steps, at 200 Hz and 50% duty cyclePulsed DC single voltage spike, 400 µs duration, at 50 Hz, calibrated by peak voltage applied at 50 to 300 V.Pulsed DC single voltage spike, 400 µs duration, at 200 Hz, calibrated by peak voltage applied at 50 to 300 V.

The different electrical treatment groups (*n* = 66) were applied randomly [16] to groups of 20 gas-stunned turkeys of around 15 weeks of age, 16–18 kgs in weight (*n* = 66 × 20 = 1320), immediately after the severance of blood vessels in the neck. The electrodes were applied to the turkey’s head before the current was switched on for a single application (≥2s). Measurement of impedance was taken 200 ms into the application of the current. A calibration signal was applied across a resistor and recorded at the start of each treatment group. 

The resultant current and voltage profiles were recorded for later analysis. The AC recordings used AC coupling and the pulsed DC recordings were made with DC coupling. The average RMS voltage and current were assessed using RMS current and voltage probes and recorded onto a Nicolet “Vision” data acquisition system. A PR 30 (LEM HEME ltd., Skelmersdale WN8 9QX, United Kingdom) current probe and a differential voltage probe (MX 9003, Metrix Electronics, Bramley, United Kingdom) were used to record current and voltage, respectively.

## 3. Results

Figure 1 shows the effect of increasing the peak voltage magnitude on impedance to current flow at 200 ms across the different waveforms and frequencies. At low voltages in all application types, impedance is relatively high, highest when DC spike at 50 Hz is used. With increasing voltage, the impedance decreases, following a non-linear pattern, until it levels off in all applications and it does not decrease any further. At 200 to 225 V and above, further decrease in the impedance is minimal in all application types.

The effect of pulsed direct currents (DC) and contribution and interaction of pulse duration, duty cycle, applied voltage, current and frequency to the effectiveness of the stun was also examined.

The ability of a voltage/waveform to result in the breakdown of the inherent high resistance of living tissue in turkeys was estimated using Figure 1. The threshold value was the minimum voltage above which an increase in voltage would not result in further reduction in impedance. In Table 2, the voltage/frequency combinations that will probably produce an effective stun in turkeys is represented.

## 4. Discussion

Electrical stunning is not the most used stunning method in slaughtering animals nowadays [9]. Gas-based stunning is more common. Electrical stunning is used in 20% of the broilers slaughtered and 99% of other poultry, like guinea fowl, ducks and geese, and in turkeys it is applied in 2.5% of the total, of which 0.5% is head only. Electrical head-only stunning is used by small scale turkey producers and constitutes a very limited number of animals annually [9,10]. Nevertheless, perhaps even because of the limited application of this method, it is important to determine effective boundaries to make sure that welfare requirements at slaughter are met in these small-scale settings. Though controlled atmosphere stunning (CAS) is currently most often used [9], and is considered a very promising method [17], it mostly is not a method of choice for small scale producers because of the cost of the systems. The same applies for water bath stunners.

Head-only electrical stunning with 400 mA delivered using 50 Hz sine wave alternating current is effective in turkeys, however neck cutting should be performed within 15 s to prevent recovery of consciousness [18]. In an overview of stunning methods including electrical stunning [19], it is stated that the current that is applied is the all-important determinant factor for the delivery of an effective stun and that it needs to be delivered using a constant current source for that reason. The magnitude of the applied voltage in breaking down the inherent high impedance of various tissues in the pathway to current flow is important [12]. The impedance of a live pig’s head was predominantly a function of the stunning voltage and decreased with increasing voltage. Following the breakdown of impedance by a high voltage application, the relationship between current and voltage would appear to match that expected from the application of Ohm’s law [12]. This is a feature of fresh or living tissue rather than the electrode/tissue interface, therefore a certain threshold voltage is needed to enable sufficient current to flow and produce an effective stun. 

The effect of increasing the peak voltage on the reduction in the initial high impedance, as shown in Figure 1, demonstrated a similar effect to that seen in pigs [12]. In our experiment, a plateau was demonstrated with the AC waveforms at approximately 212 V (peak), equivalent to 150 V RMS (0.707 × ~212), or higher. The plateau was considered to be reached when further increases in applied voltage did not result in a significant reduction in impedance, i.e., Ohm’s law was obeyed. With pigs there was a good correlation between voltage necessary to break down impedance to current flow and the voltage necessary to produce an effective stun [20]. The results shown in Figure 1, expressed as peak voltage, suggest that a lower voltage is required with turkeys, i.e., 250 V RMS at 50 Hz sine wave for pigs, and 150 V RMS at 50 Hz sine wave for turkeys. The applied voltage should be kept to a minimum to prevent accidental electrocution of the operators, hence 150 V was selected.

Using alternating current (AC), the frequency did not seem to influence the level at which the impedance levelled off. Using the pulsed direct current (DC), at the 30% duty cycle, a higher voltage was required to reduce the impedance then at the 50% duty cycle, when using 50 Hz. At 200 Hz, there was no difference between the voltage levels at which the impedance levelled off. Both showed a similar voltage as the 50 Hz, 50% duty cycle DC application; 150 V. There may be a recovery effect if low duty cycles are used in the lower frequency, but the current experiment did not investigate the physiological alterations in the tissue itself. The application of spike voltages required considerably higher voltages to reduce the impedance to comparable levels as the other applications.

High frequency stunning (1400 Hz) results in faster blood loss and a substantial reduction in haemorrhagic downgrading conditions [21,22]. Although this was determined in water bath stunning, this result provided the expectation that high frequency stunning improves carcass and meat quality in turkeys, compared to low-frequency stun-to-kill waveforms. In broilers, a study from 2017 showed no significance in the amount of damage done to the carcass depending on frequency [23]. In pigs, in addition to the effect of applied voltage on impedance, changes in frequency have also been shown to affect impedance [24].

With pigs, the effect of increasing the electrical frequency from 50 Hz to 1500 Hz was shown to increase the voltage required to deliver the same current (1.3 amps). It is suggested that this effect with pigs is likely to occur with other species e.g., turkeys.

EC Regulation 1099/2009 [2] states that electrical stunning equipment shall be fitted with a device which displays and records the details of the electrical key parameters for each animal stunned. With this level of recording, it should be possible to prescribe the electrical criteria required to promote effective electrical stunning.

Previous research has demonstrated that passing 1.0 Amp AC at 50 Hz and with 500 V for less than 0.2 s through a sheep’s head does not produce a seizure-like state characteristic for unconsciousness and insensibility to pain [25]. Consequently, the criterion proposed to predict an immediate stun with turkeys was the threshold voltage that produced a breakdown in the inherent high impedance of the turkey’s head within 0.2 s of application. Previous research has shown that the minimum current needed to stun a turkey is 400 mA [18]. However, this number was determined with sinusoidal AC at 50 Hz. The effect of varied waveforms and frequencies in this project on the minimum current to stun is unknown as the experiment was conducted on already stunned animals. A second study could therefore investigate the stunning effect of an application using the minimum voltage needed to break down impedance within 0.2 s to the minimal attainable level as a starting point.

In a study discussing variability of stunning success, particularly in relation to water bath stunning of broilers, layers and ducks, it was found that there generally is considerable variation in the impedance and success of the stun [26]. The same was found in Belgian poultry abattoirs [27].

## 5. Conclusions

The minimum voltage needed to break down the inherent resistance to current flow was assessed by applying a range of voltages head-only to turkeys with both AC and pulsed DC waveforms. Based on the need to apply sufficient voltage to break down the inherent high impedance of the turkey’s head, sinusoidal AC applied at 150 V, 50 Hz was selected for further testing.

## Figures and Tables

**Figure 1 animals-10-02427-f001:**
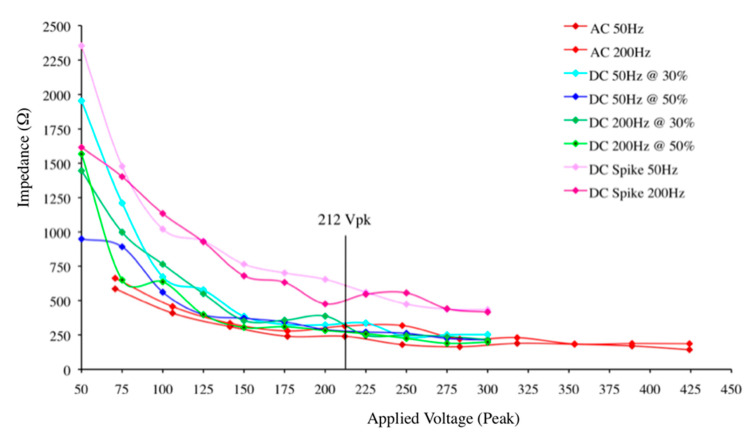
The effect of applied voltage (peak) at different waveforms on the average impedance (*n* = 20) of a turkey’s head at 200 ms after start of current flow. AC = Alternating current, DC = Direct current, DC Spike = pulsed DC single voltage spike

**Table 1 animals-10-02427-t001:** Specifications of the prototype generator.

Electrical Descriptors	Parameters
Input voltage	110/240 V AC ^1^ 50 Hz
Output voltage	50–300 V peak pulsed
Output current	1 Amp peak at 300 V with 100 Ohm
Output power	250 W
Output frequency	30–1000 Hz
Waveform	DC ^2^ pulse unidirectional

^1^ AC = Alternating current; ^2^ DC = Direct current.

**Table 2 animals-10-02427-t002:** Voltages that are expected to produce an effective stun when applied head-only to turkeys.

Waveform	Value		RMS ^1^ Equivalent
AC ^2^	50 Hz		150 V (RMS)
AC	200 Hz		150 V (RMS)
Pulsed DC ^3^	50 Hz	At 30% duty cycle	175 V (RMS)
Pulsed DC	50 Hz	At 50% duty cycle	150 V (RMS)
Pulsed DC	200 Hz	At 30% duty cycle	150 V (RMS)
Pulsed DC	200 Hz	At 50% duty cycle	150 V (RMS)
400 µs DC ^4^	50 Hz	Voltage spike	225 V (RMS)
400 µs DC	200 Hz	Voltage spike	225 V (RMS)

^1^ RMS = Root Mean Squared (0.707 × peak voltage); ^2^ AC = Alternating current; ^3^ DC = Direct current; ^4^ 400 µs DC = pulsed DC single voltage spike.

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
