# Peer review of "Head-Only Stunning of Turkeys Part 1: The Minimum Voltage Necessary to Break Down the Inherent High Resistance"

_animals, 2020, doi:10.3390/ani10122427_

Round 1

Reviewer 1 Report

The article “Head-only Stunning of Turkeys 1: The minimum Voltage necessary to break down the inherent high resistance,” describes the pre-slaughter stunning in turkeys, specifically the head-only electrical stunning. The aim of this study is clear and the conclusions correspond to the set goals. I find this article beneficial mainly due to the improvement of welfare and protection of animals during the slaughter.

I recommend this article to the publication in Animals after some minor revision: I recomend the authors to check and unify the style of writing the references section.

Author Response

Thank you very much for the favourable comments. We will cover the comment about the  style of writing the references section.

Reviewer 2 Report

The purpose and the aim of the work are described in an appropriate manner. However, I suggest some modifications.

Author Response

Thank you very much for the comments and the useful suggestions for the references. We have made the following adjustments:

Line 181: (in the revised version this is line number 196) we have checked you suggestions for the references here, and found that one does indeed confirm the findings from earlier research we already found. we have included the reference in this line. The second reference proposed here, did not find significant differences between treatments in broilers. we have included that reference in line 200.

Line 187-188: reference: on the basis of comments by another reviewer, we have removed the figures that this text is directing to. we have made changes in the text at this point.

Line 294, Reference 18: (in the revised version this is line 222) The reference is indeed incomplete, because it refers to a second publication that we have submitted to Animals, and that yet needs to be reviewed and completed. We will only be able to complete this after this second article has been accepted as well.

Lines 105-106: (in the revised version line 114-115) we have added information about slaughter age and weight of the birds.

in respect to your comment that many studies also address the effects of the stunning on the meat quality, we would like to indicate that the focus of this study was on voltage and impedance, and we have only referenced the quality aspect because we could not ignore the fact that there is a relation between these. Nevertheless, the physiological aspects and the chemical and biochemical markers mentioned were not part of this study. Therefore it does not feel appropriate to elaborate on it extensively. We have looked at the relationship between voltage level and effectiveness of the stun in a second paper that we have submitted. Again, this does not look at the physiological changes or chemical/biochemical markers.

Reviewer 3 Report

The authors assessed the most appropriate voltage in the head-only electrical stunning of turkeys in order to overcome the impedance (resistance) in the tissue of the head and therefore provide an effective stun before slaughter, that is is mandatory for humane slaughter. 
Main goal of the study in the long term is to improve the effectiveness in hand-held electrical stunning of turkeys and to reduce occurring variability in practice. The  authors were able to select sinusoidal alternate current (AC) applied at 150 volts, 50 Hz for further 
investigation.

All in all a well designed and valuable contribution to scientific literature and insight to possible improvement of animal welfare during slaughter with an electrical stunning method.

Title: Head-only Stunning of Turkeys 1: The minimum 
Voltage necessary to break down the inherent high 
resistance. 
Maybe you want to rename it to (...)stunning of turkeys Part 1:  (...).

Line 24: Please speak rather about gas stunned (gas-type? Gas concentration, duration?) than killed, since the gas stunning is a stunning method in the first place, that usually requires killing by bleeding the birds afterwards, even though it may be irreversible at the best . See also line 106.

Line 36: According to statista.com 14.8 million birds were slaughtered in the United Kingdom in 2018. https://www.statista.com/statistics/298320/turkey-slaughterings-in-the-united-kingdom-uk-by-breed/. To mention the total number per year seems to give a better feeling of how many are slaughtered together, since numbers nowadays vary not only due to the Corona pandemic. For comparison: 1.7 mio in Oct 2019, 0.6 mio in Oct 2020.

Line 42: Which is the most common electrical stunning in poultry? Head-only (simple stunning) or head-to-body (stun-to-kill) electrical stunning when using water bath stunning? A percentage and better description of the used methods for poultry/turkeys would be a valuable information, when describing the most commonly used stunning method. Please discuss why it is necessary to improve hand-held stunning of turkeys since this is mainly used in small scale slaughter premises.
 How many turkeys involved in the UK per year?

Giving the relationships via the following formular may be worth to mention and discuss in the text: Current (I) = Voltage (V) / Resistance (R).

What else, other than voltage and resistance, is essential for an effective stun when using electrical stunning? And how can an effective stun be recognised, signs of effective stun?

  • „Electricalstunning  using  a water  bath  stunner is  the most  common method  employed to  slaughter poultry under  commercial  “(Raj 1999, Poultry Science 77(12):1815-9).
  • „In small scale slaughter premises and for seasonal on-farm slaughter, birds are frequently stunned before bleeding using head-only electrical stunners.“ See also The Protection of Animals at the Time of Killing (PATK) Guidance for Poultry November 2015, Richard Griffiths, British Poultry Council
  • The authors may also want to discuss the following citation: „To date, the most effective and least aversive method of stunning birds prior to slaughter is Controlled Atmosphere Killing (CAK)“ (Shields and Raj, 2008) or Controlled Atmosphere Stun (CAS) . Could this be the future stunning method for all broilers and turkeys?

Line 44: gas stunning for example is at present a CAS (Controlled atmosphere stunning) method where smaller groups of turkeys are placed in container/tunnels with gas. Please mention used gas types/gas of choice and explain the other alternative methods a little more detailed.

Line 50: please use the head-to-body

Line 53: Please discuss the possible aid of evoked potentials/responses (EP) and EEG as an outlook for the effectiveness of assessed voltage and frequency.

In general:

Please provide an outlook on future research concerning this topic. Very interesting since the title declares the present study as part 1.

Author Response

Thank you for the favourable comments.

We have adopted your suggestion for the title and changed it into: stunning of turkeys Part 1:

Line 24: we have changed the wording into gas stunned rather than gas- killed, also in line 106 (in the revised version this is line 115).

Line 36: we have changed the wording to reflect the large variation in monthly slaughters. We have used the same reference we have originally used, which is the statistics of the Department of the Environment, Food and Rural Affairs.

Line 42: electrical head-only stunning is used by only a few small scale producers. we have added a reference for that (see line 46). Most of these are under Local Authority responsibility and there are very few records of numbers of birds slaughtered. We have adapted the text to give a bit more information. 

As to your suggestion to provide the formula for Ohm's Law, we have added that in line 59-60.

We have added a definition of effective stunning in lines 65-67.

We have included some of the references suggested in the introduction (Raj 1998, see line 44, Griffith, 2015, line 50) and Discussion (Shields and Raj, 2010, see lines 155-159).

Line 44: We have added a short sentence showing that gas stunning is done using high levels of CO2, but as this is not the scope of the paper, we think it is not approriate to discuss the several mixtures used in detail.

Line 50: In this line we do mean head-only, that is, a current is run through the head and not through head and body, as is the case in water bath stunning. therefore we have left this phrase as it was.

Line 53: in respect to your comment about the possible aid of evoked potentials/responses (EP) and EEG as an outlook for the effectiveness of assessed voltage and frequency: this is the subject of the second paper we have submitted, and it will be discussed extensively there.

Your general comment about an outlook on future research concerning this topic: I would like to recommend to the editor that this reviewer sees the second paper as well, as we continue with this work in that paper.

Round 2

Reviewer 2 Report

The requested changes have been made adequately. However, as already specified, I propose to delete the reference of the manuscript not yet published (Reference n.25).

Author Response

Thank you for your comment. We have removed the reference to the second study. We have adapted the sentence in which this was mentioned. See line 229-232.